# Early Changes in Eating Behavior Patterns and Their Relationship with Weight Outcomes in Patients Undergoing Bariatric Surgery

**DOI:** 10.3390/nu16223868

**Published:** 2024-11-13

**Authors:** Daniel Sant’Anna da Silva, Thiago Sant’Anna da Silva, Paulo Roberto Falcão Leal, Karynne Grutter Lopes, Luiz Guilherme Kraemer-Aguiar

**Affiliations:** 1Postgraduate Program in Clinical and Experimental Physiopathology (FISCLINEX), Faculty of Medical Sciences, State University of Rio de Janeiro, Rio de Janeiro 20550-013, Brazil; danielsantannadasilva@gmail.com (D.S.d.S.); tsantannadasilva@gmail.com (T.S.d.S.); prfalcaoleal@gmail.com (P.R.F.L.); kjgolrj@gmail.com (K.G.L.); 2Obesity Unit, Multiuser Clinical Research Center (CePeM), Pedro Ernesto University Hospital (HUPE), State University of Rio de Janeiro, Rio de Janeiro 20550-013, Brazil; 3General Surgery Department, Faculty of Medical Sciences, State University of Rio de Janeiro, Rio de Janeiro 20550-013, Brazil; 4Endocrinology, Department of Internal Medicine, Faculty of Medical Sciences, State University of Rio de Janeiro, Rio de Janeiro 20550-013, Brazil

**Keywords:** bariatric surgery, three-factor eating questionnaire, eating behavior, excess weight loss

## Abstract

Background/Objective: Eating behaviors (EB) correlate with weight loss after bariatric surgery (BS). Therefore, the investigation of EB could guide interventions to prevent insufficient weight outcomes. Method: A prospective cohort of 85 patients undergoing Roux-en-Y Gastric Bypass (RYGB; 84.7% female, mean age 44.7 ± 9.3 years) was included. Six months after undergoing RYGB, EB patterns, including cognitive restraint (CR), uncontrolled (UE), and emotional eating (EE), were assessed using the Three-Factor Eating Questionnaire R21 (TFEQ-R21). History, physical examination, and anthropometric assessments were collected pre- and 12 months postoperative. Patients were divided based on the percentage of excess weight loss (%EWL < 80% or ≥80%) and EB patterns were correlated with weight outcomes at 12 months. Results: The %EWL ≥ 80% group demonstrated significantly higher scores in CR and EE compared to %EWL < 80% (*p* < 0.001 and *p* = 0.01, respectively). UE scores were similar between groups (*p* = 0.41). At 12 months postoperative, the %EWL ≥ 80% group had negative correlations between CR and BMI and positive correlations between CR and EWL (rho = −0.33 and rho = 0.29; *p* = 0.04). Multiple logistic regression considering %EWL ≥ 80% as the aim outcome revealed that CR had a significant influence (OR = 1.037; *p* = 0.058), while age (OR = 0.962; *p* = 0.145), sex (OR = 2.984; *p* = 0.135), UE (OR = 0.995; *p* = 0.736), and EE (OR = 1.017; *p* = 0.296) did not. Conclusions: EBs influence outcomes after BS, and a model using TFEQ-R21 predicted them. CR six months post-surgery was the strongest predictor of higher EWL at 12 months. Further research is needed to understand the relationship between restrictive EB and BS outcomes, possibly identifying strategies to prevent disordered EB in patients with higher scores.

## 1. Introduction

Bariatric surgery effectively treats obesity with well-established benefits [1] like substantial and sustained weight loss, improved metabolic outcomes, and control of obesity-related comorbidities [2,3]. Moreover, it is a safe procedure with a low surgical mortality rate [4]. Bariatric surgery shows remarkable rates of remission of obesity-related comorbidities such as 78.5% for hypertension, 85.7% for obstructive sleep apnea, and 86.0% for type 2 diabetes mellitus (T2D) [4,5,6]. Most significantly, it decreases all-cause mortality by up to 30% over 15 years compared to clinical treatment [7], highlighting its life-saving potential.

Unfortunately, despite all the positive effects, a subgroup of patients does not achieve the expected weight outcome. Many factors may be involved in these insufficient results, and recent studies have investigated psychological factors as predictors of postoperative outcomes [8]. There is already some evidence that dysfunctional eating behaviors, especially those surveyed after surgery, have significant predictive capacity for weight outcomes after bariatric surgery [9,10]. Metabolic profiles and measures of eating disorders correlate with weight loss after surgery, possibly emphasizing the importance of targeting patients’ eating behavior patterns [11]. This evidence examined patients who were at least two years postoperative. It is possible that the study of predictive patterns early in postoperative period and their association with weight trajectory would help guide early interventions. One study showed that behavioral and psychological measures change within six months of follow-up [12], even when not significantly apparent in preoperative assessments.

Several studies support the idea that early postoperative weight loss predicts long-term weight loss [13,14,15]. One study highlights the crucial role of behavioral intervention, particularly for those with poor responses early in their weight trajectories. Weight loss in the first six months was found to have a more significant effect than clinical factors such as preoperative body mass index (BMI), age, sex, and T2D. In this study, four out of five patients who lost less than one pound (~0.45 kg) per week between the third and sixth month after bariatric surgery did not achieve the expected weight loss even when other factors were controlled [13]. This study showed that the third and sixth months after surgery are crucial. Furthermore, there are postoperative risks associated with eating behavior. Alessio et al. found a significant correlation between uncontrolled eating with weight regain and the negative aspects of quality-of-life perception two years postoperative [16]. Another study that investigated eating disorders with different tools in a more significant sample concluded that postoperative eating behavior, but not preoperative behavior, has a predictive value for the magnitude of weight loss after surgery [17].

Therefore, currently, the literature supports observing early postoperative eating behavior after bariatric surgery as a critical indicator of future body mass outcomes. We hypothesize that dysfunctional eating behaviors in the initial postoperative period are associated with insufficient body mass outcomes at 12 months. The potential implications of our findings could significantly impact the field of bariatric surgery. We aim to investigate this by correlating eating patterns obtained after bariatric surgery from a self-fulfilled validated questionnaire collected at six months and body mass measures at 12 months.

## 2. Materials and Methods

### 2.1. Patients, Study Design, and Data Collection

This prospective cross-sectional cohort study included 85 patients who underwent laparoscopic Roux-en-Y gastric bypass (RYGB) (84.7% females, aged 44.7 ± 9.3 years, preoperative BMI of 44.2 ± 5.5kg/m^2^). Patients were recruited during follow-up visits six months after RYGB in the outpatient settings. They were posteriorly allocated into two groups, according to the percentage of excess weight loss (%EWL) at 12 months (%EWL < 80%, *n* = 38 or %EWL ≥ 80%, *n* = 47) to investigate if eating behavior patterns were linked to specific weight loss outcomes. EWL ≥ 80% cutoff was used to classify those with sufficient weight loss [18,19]. The exclusion criteria were pregnancy, severe psychiatric disorders, not being able to understand/read texts in our native language, unanswered or inadequately completed questionnaires (*n* = 10), incomplete information on anthropometric data/missing the 12-month consultation (surgical data, and pre- and postoperative weight) (*n* = 10), other surgical techniques (i.e., sleeve gastrectomy) (*n* = 1), or those who did not provide informed consent. Thus, out of the 106 individuals initially recruited, 21 were excluded.

Before being eligible for the study, participants underwent a pre-participation screening, including clinical history, physical examination, and anthropometric assessment. The research protocol was explained, and all volunteers gave informed consent. Subsequently, all participants completed the Three-Factor Eating Questionnaire R21 (TFEQ-R21) at the six-month postoperative follow-up. Anthropometric data recorded before surgery and postoperative follow-up at 12 months were analyzed. Recruitment, pre-participation screening, and data collection occurred between December 2021 and February 2023. Figure 1 depicts the flow diagram of participants’ enrolment, allocation, follow-up, and data analysis. Figure 2 depicts the study design, and the procedures performed during the pre- and postoperative stages.

### 2.2. Ethical Approval

This study was approved by the Ethics Committee of the Pedro Ernesto University Hospital (CAAE: 48544621.1.0000.5259). All procedures were performed according to the principles of the Declaration of Helsinki. A signed informed consent form was obtained from each participant.

### 2.3. Anthropometric Assessment

Height and body mass were measured using a stadiometer and calibrated electronic scale (Welmy™ W300A, São Paulo, SP, Brazil). BMI was calculated as the ratio of body mass to height squared (kg/m^2^). Preoperative weight and minimum postoperative weight were obtained in the medical record. %EWL was calculated according to the following equation: %EWL = [(preoperative weight—minimum postoperative weight)/(preoperative weight—ideal weight; considering BMI of 25 kg/m^2^ as the ideal target)] × 100.

### 2.4. Three-Factor Eating Questionnaire R21

The TFEQ-R21 was used for sample characterization. It is a self-administered psychometric questionnaire validated in Portuguese. It measures dysfunctional eating behavior into three domains: (1) cognitive restraint scale, which is the conscious restriction of food intake to influence body weight and body shape; (2) uncontrolled eating scale, which is a tendency to lose control over eating when feeling hungry or exposed to external stimuli; and (3) emotional eating scale, which assesses the propensity to overeat when exposed to negative mood states, such as loneliness, anxiety, or depression. Twenty-one questions on 4-point scales for items 1 to 20 (“totally true”, “mostly true”, “mostly false” and “totally false”) or an 8-point scale for item 21 (a scale from 1 to 8 with 1 = “eat whatever and whenever I want” and 8 = “constantly limit food intake, never giving in”) composes its structure [20,21].

### 2.5. Statistical Analysis

Data normality was assessed using the Shapiro−Wilk test. Between-group differences were determined using the unpaired Student t, Mann−Whitney, or Chi-square tests. Results were expressed as mean ± standard deviation, median [percentiles 25–75], or frequency (*n*, %). Pearson’s or Spearman’s rho correlation coefficients were calculated to investigate the associations between three domains correlated with dysfunctional eating behavior vs. demographic characteristics and bariatric surgery data, as appropriate. Multiple logistic regression analysis estimated the influence of eating behavior patterns, age, and gender with %EWL at 12 months. All calculations were performed using Jamovi version 2.3 statistical package (The Jamovi project, Sidney, Australia) and NCSS^TM^ statistical software (Version 1, LLC, Kaysville, UT, USA). Statistical significance was set at *p* < 0.05.

## 3. Results

Table 1 presents the demographic characteristics, bariatric surgery data, and eating behavior patterns of the pooled sample and the study groups. Our sample was mainly composed of females with obesity grade 3 before surgery and a mean EWL of 84.1% after 12 months. The group that had poorer results 12 months postoperative (EWL < 80%) was older (*p* = 0.03) and had higher weight and BMI than the other group (*p* < 0.001 for both variables) before surgery. Those with EWL ≥ 80% exhibited higher cognitive restraint and emotional eating scores than those with EWL < 80% (*p* < 0.001 and *p* = 0.01), while uncontrolled eating scores were similar between groups (*p* = 0.41).

Associations between the postoperative data and eating behavior patterns are depicted in Table 2. Of note, in the pooled sample, we noticed one positive correlation for cognitive restraint vs. EWL at 12 months (*p* = 0.02) and three negative correlations for cognitive restraint vs. weight/EWL at 12 months (*p* < 0.001 and *p* = 0.01, respectively), and emotional eating vs. age (*p* < 0.001). The cognitive restraint vs. BMI and EWL at 12 months were possibly influenced by the results from the %EWL ≥ 80% group (*p* = 0.04 for both variables).

A model composed of eating behavior patterns, gender, and age was used to estimate their influence on EWL at 12 months (the *p*−value of the model was 0.049). Overall, considering age and eating behavior patterns, only cognitive restraint had higher odds of being in the %EWL ≥ 80% group (OR = 1.037; *p* = 0.058), compared with age (OR = 0.962; *p* = 0.145), gender (OR = 2.984; *p* = 0.135), uncontrolled eating (OR = 0.995; *p* = 0.736), and emotional eating (OR = 1.017; *p* = 0.296).

## 4. Discussion

In this comprehensive study, we thoroughly investigated the relationship between the indicators of the TFEQ-R21, sex, age, excess weight loss at 12 months, and the outcome of the percentage of EWL at 12 months (%EWL 12 m ≥ 80%) in patients who were submitted to Roux-en-Y Gastric Bypass (RYGB). We hypothesized that high scores in TFEQ-R21 indicators would be associated with lower-than-expected weight loss. However, surprisingly, the results do not support our initial hypothesis since we did not find any evidence that the indicators of uncontrolled eating (UE) and or emotional eating (EE) are related to an unfavorable EWL outcome 12 months postoperative. Our meticulous approach to this study ensures the validity of our findings and indicates that the clinical effects of bariatric surgery may outweigh unfavorable eating behaviors regarding weight loss.

A crucial aspect herein identified was the association between cognitive restraint (CR) and successful weight outcomes since patients with higher CR scores were more likely to achieve an EWL ≥ 80% at 12 months postoperative (Table 2). The relationship between higher scores in CR and EWL has already been described, but specifically in patients who had undergone sleeve gastrectomy [22]. Our finding, tested in patients after RYGB, suggests that the tendency to control food intake related to weight and body shape concerns, characteristic of CR behavior, plays a significant role in a successful weight outcome after bariatric surgery. Based on this observation, patients with low CR scores should be closely monitored for the risk of achieving lower-than-expected excess weight loss.

It is important to note that TFEQ-R21 used in this study was not specifically developed for bariatric surgery patients. It may lead to varied interpretations of specific questions, such as ‘I deliberately take small helpings to control my weight’, potentially reflecting appropriate therapeutic adherence. Additionally, the association between high CR scores and an EWL ≥ 80% outcome may foster a dynamic that increases the risk of developing eating disorders [23,24]. Patients displaying more intense CR behavior might be positively reinforced by their weight loss success, potentially perpetuating cycles of restrictive patterns [25]. Given that the literature supports the emergence of eating disorders post-bariatric surgery [9,10], it is crucial to monitor patients exhibiting any dysfunctional eating behaviors closely.

Worse outcomes in the older group may be explained by physiological changes in body composition over time, with a tendency towards decreasing lean mass and increasing body fat. The selection bias of this group justifies the poorer results associated with higher initial weight and BMI, as these patients naturally have a more severe disease phenotype due to metabolic, genetic, or behavioral factors. Furthermore, the methodology employed to calculate EWL requires greater absolute weight loss to achieve the same relative weight loss compared to patients with lower initial BMIs.

Regarding the positive relationship between EE and EWL > 80%, in 2011, Natacci and Ferreira Júnior described a positive correlation between CR and EE [21]. They inferred that CR may render individuals vulnerable to EE, increasing their reactivity to food-related sensory or cognitive cues. Our results also showed that a model including some TFEQ-R21 indicators along with sex and age has a moderate to good predictive ability for a better outcome at 12 months postoperative. This aspect aligns with the literature emphasizing the importance of closely monitoring eating behavior in the postoperative period since initial weight loss is a significant predictor of long-term weight loss. These findings underscore the relevance of considering clinical factors and patient eating behavior in evaluating postoperative success.

Some limitations of our study, like the sample size and the lack of longitudinal assessments of eating behavior, should to be mentioned. Moreover, we did not monitor medication use, and those using medications that influence behavior and/or body weight could have biased our results. However, we should reemphasize that during the first year postoperative, a “honeymoon period” is established, during which most patients require minimal use of medications. Additionally, we opted to not investigate the dietary composition of the volunteers since the subdivisions of the groups according to dietary patterns would possibly create excessive heterogeneity and hinder the clinical applicability of the questionnaire. Specific adherence questionnaires were also not used, as the existing questionnaire already includes several items related to dietary adherence. Of note, we highlight that adherence to exercise was not assessed. It is well known that physical activity influences bariatric surgery outcomes in both the short and long term [26].

## 5. Conclusions

Eating behavior plays a significant role in the success of weight trajectories after bariatric surgery. A model that incorporates TFEQ-R21 indicators, along with sex and age, can help predict better weight outcomes. CR behavior observed six months after RYGB was the indicator that correlated most with higher EWL at 12 months. Further research is needed to better understand how restrictive eating behaviors evolve in parallel or as a function of the weight loss trajectory after bariatric surgery. Moreover, it may help identify early intervention strategies to prevent the development of eating disorders in patients with a high CR score. It is our responsibility to ensure close monitoring of these patients. Additional studies will help us to understand the complex interactions between eating behavior and weight loss after bariatric surgery, particularly those assessing the risk of early post-surgical eating disorders.

## Figures and Tables

**Figure 1 nutrients-16-03868-f001:**
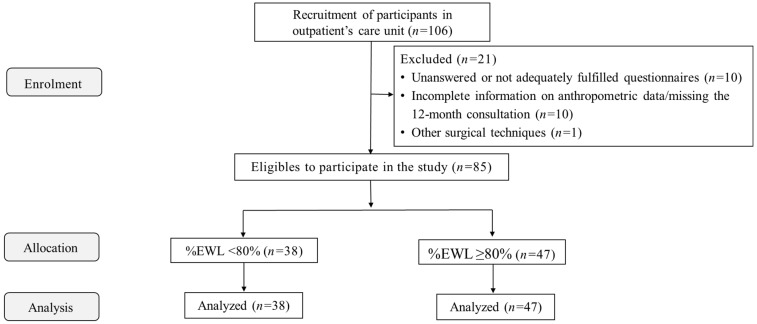
Flow diagram of participants’ enrolment, allocation, follow-up, and data analysis. EWL—excess weight loss; %EWL < 80—patients with a EWL at 12 months <80%; %EWL ≥ 80%—patients with a EWL at 12 months ≥80%; TFEQ-R21—Three-Factor Eating Questionnaire R21.

**Figure 2 nutrients-16-03868-f002:**
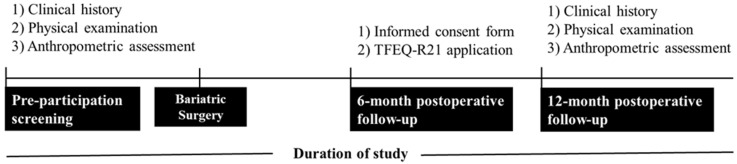
Study design and the procedures performed during the pre- and postoperative stages.

**Table 1 nutrients-16-03868-t001:** Demographic characteristics, bariatric surgery data, and eating behavior patterns of study groups.

Variable	Pooled Sample (*n =* 85)	%EWL < 80% (*n =* 38)	%EWL ≥ 80% (*n =* 47)
Demographic characteristics			
Age, years	44.7 ± 9.3	46.8 ± 9.2	43.1 ± 9.1 *
Female (*n*, %)	72 (84.7)	34 (89.5)	38 (80.9)
Preoperative weight (kg)	116 [105.0–128.7]	122.8 [109.5–134.9]	113.7 [101–122.5] *
Preoperative BMI (kg/m^2^)	44.3 ± 5.5	46.9 ± 5.1	42.0 ± 4.8 *
Bariatric surgery data			
BMI at 12 months (kg/m^2^)	28.3 [25.9–30.7]	30.9 [29.9–34.4]	26.5 [23.8–27.3] *
EWL at 12 months (%)	84.1 ± 18.2	67.9 ± 6.9	97.2 ± 13.2 *
Eating behavior patterns (scores)			
Cognitive restraint	37.0 [25.9–44.4]	35.2 [22.2–40.7]	37.0 [29.6–48.1] *
Uncontrolled eating	38.9 [22.2–55.5]	30.5 [22.2–56.9]	44.4 [22.2–55.5]
Emotional eating	11.1 [0–33.3]	11.1 [0–33.3]	16.6 [5.5–38.9] *

EWL—excess weight loss; %EWL < 80—patients with a EWL at 12 months <80%; %EWL ≥ 80%—patients with a EWL at 12 months ≥80%; BMI—body mass index. *p*−values are results from between−group comparisons (* *p* < 0.05); data presented as mean ± standard deviation, median [percentiles 25–75] or *n*, %.

**Table 2 nutrients-16-03868-t002:** Correlation coefficients between eating behavior patterns, age, and bariatric surgery data in the pooled sample and according to groups.

Variable	Pooled Sample (*n =* 85)			%EWL < 80% (*n =* 38)			%EWL ≥ 80% (*n =* 47)		
	CR	UE	EE	CR	UE	EE	CR	UE	EE
Age (Years)	−0.09	−0.10	−0.26 ***	−0.02	−0.12	−0.22	−0.07	−0.08	−0.26
Preoperatory weight (kg)	−0.21	−0.008	−0.15	−0.01	−0.07	0.02	−0.20	0.06	−0.22
Preoperatory BMI (kg/m^2^)	−0.19	−0.05	−0.10	0.01	−0.13	−0.13	−0.21	0.04	0.01
Weight at 12 months (kg)	−0.27 ***	0.01	−0.10	−0.12	−0.03	0.10	−0.25	0.11	−0.14
BMI at 12 months (kg/m^2^)	−0.25 **	−0.02	−0.06	0.04	−0.09	−0.04	−0.33 *	0.10	0.21
EWL at 12 months (%)	0.27 *	−0.02	−0.003	−0.08	−0.01	−0.08	0.29 *	−0.11	−0.24

CR—cognitive restraint; UE—uncontrolled eating; EE—emotional eating; EWL—excess weight loss; %EWL < 80—patients with a EWL at 12 months <80%; %EWL ≥ 80%—patients with a EWL at 12 months ≥80%; BMI—body mass index. *p*−values * < 0.05, ** < 0.01, and *** < 0.001.

## Data Availability

The original contributions presented in the study are included in the article, further inquiries can be directed to the corresponding author.

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
