# Peer review of "Early Changes in Eating Behavior Patterns and Their Relationship with Weight Outcomes in Patients Undergoing Bariatric Surgery"

_nutrients, 2024, doi:10.3390/nu16223868_

Round 1

Reviewer 1 Report

Comments and Suggestions for Authors

The authors aimed to test the behavioral eating habits on the weight-loss after 12 mo. bariatric surgery in severe obese females using the Three-Factor Eating Questionnaire R21 (TFEQ-R21). This is a prospectic study; the methods are appropriated and statistical analyses robust and appropriate. The conclusions are supported by the results.

Some major  points:

1. All the recruited patients are females, we suggest to change the title accordingly;

2. The authors recruited the patients between 2021 and february 2023; have the authors tested the effects of COVID-19 pandemic disconfort on  EE? 

3. No data have been provided on the diet composition and adhesion grade after the bariatric surgery;

4. Similarly, no data have been provided on the levels of sedentary in recruited patients. Adapted physicl activity programs have been recently applied to the patients before and after bariatric surgery to improve the weight-loss and contribute to reach the weight -loss and maintain . The authors must discuss also this aspect .

Author Response

Rio de Janeiro, November 10th, 2024.

Dear Editors and Reviewers,

We appreciate the comments and suggestions made by reviewers. The manuscript has been revised, and itemized responses are now available. The answers are at the end of each reviewer's comment (underlined). We have performed manuscript changes which are now re-submitted with tracked changes exposed. The manuscript improved due to the suggestions made, and we hope it now receives your approval for publication.

Yours Sincerely,

L.G. Kraemer-Aguiar on behalf of all authors

COMMENTS AND SUGGESTIONS FOR AUTHORS

Reviewer #1

The authors aimed to test the behavioral eating habits on the weight-loss after 12 mo. bariatric surgery in severe obese females using the Three-Factor Eating Questionnaire R21 (TFEQ-R21). This is a prospectic study; the methods are appropriated and statistical analyses robust and appropriate. The conclusions are supported by the results.

Some major points:

(reviewer #1, comment #1): All the recruited patients are females, we suggest to change the title accordingly.

Answer – (reviewer #1, comment #1): As described in line 159, “Our sample was mainly composed of females with obesity grade 3 before surgery”. Gender information can be found in Table 1; we had 85 volunteers recruited, with 72 being female, corresponding to 84.7% of the sample. Although the majority are females, we had male volunteers, and we believe it is better to keep the title unchanged to avoid misinterpretation of our data.

(reviewer #1, comment #2): The authors recruited the patients between 2021 and february 2023; have the authors tested the effects of COVID-19 pandemic disconfort on  EE? 

Answer – (reviewer #1, comment #2): We appreciated this comment. No questions or surveys specifically addressing the effects of the COVID-19 pandemic were conducted. We believe that this variable may have been equally distributed across the sample, thus not significantly impacting or biasing the results.

(reviewer #1, comment #3): No data have been provided on the diet composition and adhesion grade after the bariatric surgery;

Answer – (reviewer #1, comment #3): We appreciated the reviewer´s comment, but regarding early postoperative weight outcomes after bariatric surgery, we aimed to evaluate the questionnaire's ability to differentiate patients with better or worse weight outcomes. Attempting to distinguish dietary composition within the group further would create additional subgroups, making the sample overly heterogeneous while possibly diminishing the clinical applicability of the questionnaire herein tested. Regarding your comment on dietary adherence, although not directly programmed to test it, in our view, the questionnaire indirectly assesses it since it includes items related to eating behavior, impulse control, and restriction capacity, even though we recognized it as a limitation of the study.

(reviewer #1, comment #4): Similarly, no data have been provided on the levels of sedentary in recruited patients. Adapted physicl activity programs have been recently applied to the patients before and after bariatric surgery to improve the weight-loss and contribute to reach the weight -loss and maintain . The authors must discuss also this aspect

Answer – (reviewer #1, comment #4): Evaluating the impact of exercise on weight outcomes following bariatric surgery was not within the initial scope of the study, even knowing that it potentially influences short- and long-term results. Therefore, we agree that adding it as a limitation is essential, highlighting that this variable was not assessed. We have added the reference below to this section.

  1. Bellicha A, van Baak MA, Battista F, Beaulieu K, Blundell JE, Busetto L, Carraça EV, Dicker D, Encantado J, Ermolao A, Farpour-Lambert N, Pramono A, Woodward E, Oppert JM. Effect of exercise training before and after bariatric surgery: A systematic review and meta-analysis. Obes Rev. 2021 Jul;22 Suppl 4(Suppl 4):e13296. doi: 10.1111/obr.13296. Epub 2021 Jun 3. PMID: 34080281; PMCID: PMC8365633.

Reviewer #2

Thank you for the opportunity to review the original article entitled ‘Early changes in eating behavior patterns and their relationship with weight outcomes in patients undergoing bariatric surgery’ that aims to evaluate the effect of emotional drivers of eating behavior on excess weight loss on a group of 85 patients who undergone bariatric surgery in the previous year. This is vital in weight regain after different weight loss interventions (lifestyle optimization, drugs, or bariatric surgery) with practical implications. This aspect is well described in the Introduction section and provides the study's rationale. In the material and methods section, the authors provide a clear depiction of the inclusion and exclusion criteria and the study’s design. These are well described in the figures included in this section. The results are easy to understand, and relevant for the study’s purpose. The tables are relevant to the text of the article included in this section.

(reviewer #2, comment #1): The discussion section is well organized, and the authors comment on each individual aspect of their findings. I suggest including references for the commentary included in lines 216-219 and 228-230 that are relevant to the results.

Answer – (reviewer #2, comment #1): We appreciated your review, which immensely helped us to consider essential details relevant to the clinical reasoning of the issue. We will add some references for the cited section. Regarding the reference for lines 216-219, we will include a reference found in the literature, but we will provide a brief explanation in the original manuscript to incorporate it.

 References:

  1. Bakr AA, Fahmy MH, Elward AS, Balamoun HA, Ibrahim MY, Eldahdoh RM. Analysis of medium-term weight regain 5 years after laparoscopic sleeve gastrectomy. Obes Surg. 2019 Nov;29(11):3508-3513. doi: 10.1007/s11695-019-04009-w.
  2. Klapsas M, Hindle A. Patients' Pre and Post-Bariatric Surgery Experience of Dieting Behaviours: Implications for Early Intervention. Obes Surg. 2023; 33(9):2702-10.doi:10.1007/s11695-023-06689-x (already mentioned in our article).
  3. Watson C, Riazi A, Ratcliffe D. Exploring the experiences of women who develop restrictive eating behaviours after bariatric surgery. Obes Surg. 2020 Jun;30(6):2131-2139. doi: 10.1007/s11695-020-04424-4.

(reviewer #2, comment #2): The conclusion section provides statements according to the study’s result and further direction important for the research.

Answer – (reviewer #2, comment #2): Thank you for your comment. We feel grateful for noting that our work was appreciated and may add knowledge to this area of interest. After your comments above, we opted to change the conclusion, expressing that our data was investigated, especially in those who had undergone RYGB.

Reviewer 2 Report

Comments and Suggestions for Authors

Thank you for the opportunity to review the original article entitled ‘Early changes in eating behavior patterns and their relationship with weight outcomes in patients undergoing bariatric surgery’ that aims to evaluate the effect of emotional drivers of eating behavior on excess weight loss on a group of 85 patients who undergone bariatric surgery in the previous year. This is vital in weight regain after different weight loss interventions (lifestyle optimization, drugs, or bariatric surgery) with practical implications. This aspect is well described in the Introduction section and provides the study's rationale.

In the Material and Methods section, the authors provide a clear depiction of the inclusion and exclusion criteria and the study’s design. These are well described in the figures included in this section.

The results are easy to understand, and relevant for the study’s purpose. The tables are relevant to the text of the article included in this section.

The discussion section is well organized, and the authors comment on each individual aspect of their findings. I suggest including references for the commentary included in lines 216-219 and 228-230 that are relevant to the results.

The conclusion section provides statements according to the study’s result and further direction important for the research.

Kind regards

Author Response

(The authors gave the same response as above.)
